# Subspace Clustering via Robust Self-Supervised Convolutional Neural Network

## Abstract

Deep subspace clustering (SC) algorithms recently gained attention due to their ability to successfully handle nonlinearities in data. However, the insufficient capability of existing SC methods to deal with data corruption of unknown (arbitrary) origin hinders their generalization ability and capability to address real-world data clustering problems. This paper proposes the robust formulation of the self-supervised convolutional subspace clustering network ($S^2$ConvSCN) that incorporates the fully connected (FC) layer and, with an additional spectral clustering module, is capable of estimating the clustering error without using the ground truth. Robustness to data corruptions is achieved by using the correntropy induced metric (CIM) of the error that also enhanced the generalization capability of the network. The experimental finding showed that CIM reduces sensitivity to overfitting during the learning process and yields better clustering results. In a truly unsupervised training environment, Robust $S^2$ConvSCN outperforms its baseline version by a significant amount for both seen and unseen data on four well-known datasets.

## 1 Introduction

Subspace clustering approaches have achieved encouraging performance when compared with the clustering algorithms that rely on proximity measures between data points. The main idea behind the subspace model is that the data can be drawn from low-dimensional subspaces which are embedded in a high-dimensional ambient space (Lodhi & Bajwa, 2018). Grouping such data associated with respective subspaces is known as the subspace clustering (Vidal, 2011). That is, each low-dimensional subspace corresponds to a class or category. Up to now, two main approaches for recovering low-dimensional subspaces are developed: models that are based on the self-representation property, and non-linear generalization of subspace clustering called *union of subspaces* (UoS) (Lodhi & Bajwa, 2018; Lu & Do, 2008; Wu & Bajwa, 2014; 2015). UoS algorithms are out of the scope of this work. Self-representation subspace clustering is achieved in two steps: (i) learning representation matrix $\mathbf{C}$ from data $\mathbf{X}$ and building corresponding affinity matrix $\mathbf{A} = |\mathbf{C}| + |\mathbf{C}^T|$; (ii) clustering the data into $k$ clusters by grouping the eigenvectors of the graph Laplacian matrix that correspond with the leading $k$ eigenvalues. This second step is known as spectral clustering (Ng et al., 2002; Von Luxburg, 2007). Owning to the presumed subspace structure, the data points obey the self-expressiveness or self-representation property (Elhamifar & Vidal, 2013; Peng et al., 2016b; Liu et al., 2012; Li & Vidal, 2016; Favaro et al., 2011). In other words, each data point can be represented as a linear combination of other points in a dataset: $\mathbf{X}=\mathbf{XC}$.

The self-representation approach is facing serious limitations regarding real-world datasets. One limitation relates to the linearity assumption because in a wide range of applications samples lie in nonlinear subspaces, e.g. face images acquired under non-uniform illumination and different poses (Ji et al., 2017). Standard practice for handling data from nonlinear manifolds is to use the kernel trick on samples mapped implicitly into high dimensional space. Therein, samples better conform to linear subspaces (Patel et al., 2013; Patel & Vidal, 2014; Xiao et al., 2015; Brbić & Kopriva, 2018). However, identifying an appropriate kernel function for a given data set is quite a difficult task (Zhang et al., 2019b). The second limitation of existing deep SC methods relates to their assumption that the origin of data corruption is known, in which case the proper error model can be employed. In real-word applications origin of data corruption is unknown. That can severely harm the algorithm's learning process if the non-robust loss function is used. Furthermore, validation

(i.e. stopping of the learning process) in most of the deep SC methods often requires access to the ground-truth labels. That stands for violation of the basic principle of unsupervised machine learning and yields the overly-optimistic results. Dataset size is also a limitation when it comes to memory requirements. Since the self-representation subspace clustering is based on building the affinity matrix, memory complexity increases as the square of the dataset size. However, the latter limitation is not in the main focus of this work.

Motivated by the exceptional ability of deep neural networks to capture complex underlying structures of data and learn discriminative features for clustering (Hinton & Salakhutdinov, 2006; Dilokthanakul et al., 2016; Ghasedi Dizaji et al., 2017; Tian et al., 2014; Xie et al., 2016), deep subspace clustering approaches emerged recently (Ji et al., 2017; Abavisani & Patel, 2018; Peng et al., 2016a; Yang et al., 2019; Zhou et al., 2018; Ji et al., 2019b; Peng et al., 2018; 2017; Zhou et al., 2019; Zhang et al., 2019a; Kheirandishfard et al., 2020). In particular, it is shown that convolutional neural networks (CNNs), when applied to images of different classes, can learn features that lie in a UoS (Lezama et al., 2018). Mostly, the base of the recently developed deep subspace-clustering networks is convolutional autoencoder. It is an end-to-end fully convolutional network that is based on the minimization of the reconstruction error. Together, the autoencoder and an additional self-expression (SE) module are forming a Deep subspace clustering network (DSCNet) (Ji et al., 2017). Hence, the total loss function of DSCNet is composed of reconstruction loss and SE model loss. That is, during the learning process the clustering quality is not taken into account. Self-supervised convolutional SC network (S$^2$ConvSCN) (Zhang et al., 2019a) addressed this issue through the addition of a fully connected layer (FC) module and a spectral clustering module that, respectively, generate soft- and pseudo-labels. Dual self-supervision is achieved by forcing these two modules to converge towards consensus. Related accumulated loss, therefore, participates in enhancing the self-representation matrix and the quality of features extracted in the encoder layer. The architecture of S$^2$ConvSCN has a possibility of direct classification once the learning process is completed. A trained encoder and the FC module can make a new network that can directly classify unseen data, also known as an out-of-sample problem. However, while this network can be validated and compared with other algorithms on a separate data set, such an ablation study was not completed. Furthermore, the main disadvantage of the DSCNet architecture, and indirectly S$^2$ConvSCN, is that the network training is stopped when the accuracy is highest (Ji et al., 2019a). First, it is a direct violation of the unsupervised learning principle as the ground-truth labels are exposed. Second, the reported performance (Zhang et al., 2019a; Ji et al., 2017) is overly-optimistic and can not be compared to other algorithms. Also, as mentioned in (Haeffele et al., 2020), most self-expressive based deep subspace clustering models suffer from the need of post-processing the self-representation matrix. Compared to the baseline model, we significantly reduced the post-processing while maintaining the noise-free matrix.

Mentioned research problems led to three main contributions of proposed Robust S$^2$ConvSCN:

- robustness to errors of the unknown (arbitrary) origin is achieved by using the correntropy induced metric (CIM) in the self-expression loss,

- the network is trained using the early-stopping method while monitoring only the accumulated loss,

- thanks to correntropy based loss function the training process is less sensitive to data corruptions which enables the network to generalize better.

This study has, also, three side-contributions:

- the performance of models is estimated using the unseen (out-of-sample) data,

- block-diagonal regularization of self-representation matrix is integrated into the gradient descent learning process,

- post-processing of self-representation matrix is reduced to a significant extent.

A complete head to head comparison of the baseline S$^2$ConvSCN model and our robust approach can be seen in Figure 1.

| | Self-expressive layer | Early stopping | Evaluation | Robustness |
|---|---|---|---|---|
| **Baseline approach:** | $\mathrm{MSE} + \ell_2$ | Highest accuracy - requires access to ground-truth labels | On in-sample labels used for hyperparameter optimization | To additive white Gaussian noise |
| **Our approach:** | $\mathrm{CIM} + \ell_2$ $\mathrm{CIM} + \mathrm{BD}$ $\frac{1}{2}\|\mathbf{C} - \mathbf{A}\|^2$ | Loss decrease plateau - based on label-free relative error criterion | On out-of-sample labels previously unseen by the model | To data corruption of unknown (arbitrary) origin |

Figure 1: Key differences between the baseline S$^2$ConvSCN and proposed robust version. Unlike the baseline S$^2$ConvSCN, robust version learns in a truly unsupervised (label-free) manner. The last regularization in the self-expressive layer stands for symmetric loss introduced in Equation (10).

## 2 BACKGROUND AND RELATED WORK

### 2.1 MAIN NOTATIONS AND DEFINITIONS

Throughout this paper, matrices are represented with bold capital symbols and vectors with bold lower-case symbols. $\mathbf{X} \in \mathbb{R}^{d \times N}$ represents data matrix comprised of $N$ data samples with dimensionality $d$. $\left\{\mathbf{H}_i^{(l)}\right\}_{i=1}^{m^{(l)}}$ represent feature maps produced at the output of layer $l-1$. Thus, $\mathbf{H}^{(0)} = \mathbf{X}$ and $\mathbf{H}^{(L)} = \hat{\mathbf{X}}$. $\hat{\mathbf{X}}$ represents the output of the decoder and $L$ represents number of convolutional layers in the autoencoder. $\left\{\mathbf{w}_i^{(l)}\right\}_{i=1}^{m^{(l)}}$ stand for a set of filters with associated biases $\left\{\mathbf{b}_i^{(l)}\right\}_{i=1}^{m^{(l)}}$ that form a convolutional layer $l = 1, \ldots, L$. $\mathbf{z}_n = \left[\mathbf{h}_1^{(L/2)}(:) \ldots \mathbf{h}_{m^{(L/2)}}^{(L/2)}(:)\right]^T \in \mathbb{R}^{\hat{d} \times 1}$ stands for feature vector comprised of vectorized and concatenated feature maps, with $\hat{d}$ extracted features, in the top layer $\frac{L}{2}$ (encoder output) representing input sample $\mathbf{x}_n, n = 1, \ldots, N$. $\mathbf{C} \in \mathbb{R}^{N \times N}$ stands for representation matrix in self-expressive model $\mathbf{Z} = \mathbf{ZC}$. $\mathbf{A} = |\mathbf{C}| + |\mathbf{C}^T|$ is the affinity matrix and $\mathbf{L} = \mathbf{D}^{-\frac{1}{2}} \mathbf{A} \mathbf{D}^{\frac{1}{2}}$ is corresponding graph Laplacian matrix. $\mathbf{D}$ is diagonal degree matrix such that $\mathbf{D}_{ii} = \sum_{j=1}^N \mathbf{A}_{ij}$. $\|\mathbf{X}\|_F = \sqrt{\sum_{i,j=1}^N x_{ij}^2}$ is the Frobenius norm of matrix $\mathbf{X}$. $\ell_p(\mathbf{x}) = \|\mathbf{x}\|_p = (\sum_{i=1}^d \|x_i\|^p)^{1/p}$, $0 < p \leq 1$ is the $\ell_p$ norm of $\mathbf{x}$. $\ell_0(\mathbf{x}) = \|\mathbf{x}\|_0 = \#\{x_i \neq 0, \ i = 1, \ldots, d\}$, where # denotes the cardinality function, is $\ell_0$ quasi norm of $\mathbf{x}$. The $S_p$, $0 < p \leq 1$, Schatten norms of matrix $\mathbf{X}$ are defined as the corresponding $\ell_p$ norms of the vector of singular values of $\mathbf{X}$, i.e. $S_p(\mathbf{X}) = \|\sigma(\mathbf{X})\|_p$ where $\sigma(\mathbf{X})$ stands for the vector of singular values of $\mathbf{X}$. Depending on the context, $\mathbf{0}$ represents matrix/vector of all zeros and $\mathbf{1}$ represents matrix/vector of all ones.

Grouping the data according to the linear subspaces they are drawn from is known as subspace clustering (Vidal, 2011). The problem is formally defined in:

**Definition 1.** Let $\mathbf{X} = [\mathbf{X}_1, \ldots, \mathbf{X}_k]$ be a set of sample vectors drawn from a union of $k$ subspaces in $\mathbb{R}^d$, $\cup_{i=1}^k \{S_i\}$, of dimensions $d_i \ll \min\{d, N\}$, for $i = 1, \ldots, k$. Let $\mathbf{X}_i$ be a collection of $N_i$ samples drawn from subspace $S_i$, $N = \sum_{i=1}^k N_i$. The problem of subspace clustering is to segment samples into the subspaces they are drawn from. Throughout this paper, as it is the case in the majority of other papers, we have assumed that number of clusters $k$ is known *a priori*.

### 2.2 APPROACHES TO SUBSPACE CLUSTERING

Usually, processes that operate in different modes generate data in real-world scenarios. Each mode models such data as lying on a subspace, while the whole process, thus, generates data lying on a union of subspaces (UoS) (Lodhi & Bajwa, 2018). The alternative to the UoS model is the self-representation based subspace model. It implies that every sample from the dataset can be represented as a linear combination of other samples from the same cluster. While shallow models

directly optimize such a self-representation matrix, their deep counterparts train the whole network to better extract features from the raw data and achieve representation linearity.

Many approaches to deep subspace clustering are based on the introduction of the self-representation in the feature space (Abavisani & Patel, 2018; Ji et al., 2017; Peng et al., 2016a; Zhou et al., 2018; 2019; Zhang et al., 2019a; Kheirandishfard et al., 2020; Zhang et al., 2020). However, one weakness of self-expressive deep subspace clustering models is that their perfomance mainly depends on the self-representation matrix. Thus, elimination of the noise is done by post-processing (Haeffele et al., 2020). It appears in many cases that from the final performance point of view the post-processing matters more than depth of the network. By the virtue of self-representation property, improvements of the shallow subspace clustering methods are of direct relevance to their deep counterparts. The subspace clustering task is accomplished through (i) learning the representation matrix $\mathbf{C}$ from data $\mathbf{X}$, and (ii) clustering the data into $k$ clusters by grouping the eigenvectors of the graph Laplacian matrix $\mathbf{L}$ that correspond with the $k$ leading eigenvalues. This second step is known as spectral clustering (Ng et al., 2002; Von Luxburg, 2007). Low-rank (Liu et al., 2012; Favaro et al., 2011) and sparse models (Elhamifar & Vidal, 2013) are one of the commonly used algorithms to solve SC clustering problem. They aim to learn the low-rank and sparse representation matrix by solving the following optimization problem (Li & Vidal, 2016):

$$\min_{\mathbf{C}} \lambda \|\mathbf{C}\|_{S_p}^p + \tau \|\mathbf{C}\|_p^p \ s.t. \ \mathbf{Z} = \mathbf{ZC}, \ diag(\mathbf{C}) = \mathbf{0} \tag{1}$$

where $\lambda$ and $\tau$ are nonnegative regularization constants. If number of layers $L = 0$ problem (1) is related to shallow subspace clustering. Constraint $diag(\mathbf{C}) = \mathbf{0}$ is necessary to prevent sparseness regularized optimization algorithms to converge towards trivial solution where each data point represents itself. This constraint is not necessary for problem constrained only by low-rank. When data samples are contaminated with additive white Gaussian noise (AWGN) problem (1) becomes:

$$\min_{\mathbf{C}} \|\mathbf{E}\|_F^2 + \lambda \|\mathbf{C}\|_{S_p}^p + \tau \|\mathbf{C}\|_p^p \ s.t. \ diag(\mathbf{C}) = \mathbf{0} \tag{2}$$

where $\mathbf{E}$ stands for the modelling error (noise):

$$\mathbf{E} = \mathbf{Z} - \mathbf{ZC}. \tag{3}$$

Alternatively, square of the Frobenius norm of $\mathbf{C}$ is used for regularization (Lu et al., 2012):

$$\min_{\mathbf{C}} \|\mathbf{E}\|_F^2 + \lambda \|\mathbf{C}\|_F^2 \tag{4}$$

Objective (4) is used also in the self-expression module of the S$^2$ConvSCN in (Zhang et al., 2019a). As seen from (2) and (4), the MSE measure for discrepancy between $\mathbf{Z}$ and its self-representation $\mathbf{ZC}$ is justified only for the contamination by the AWGN. For sample-specific corruptions (outliers) the proper norm is $\|\mathbf{E}\|_{2,1}$ while for large random corruptions the proper choice is $\|\mathbf{E}\|_1$ (Liu et al., 2012). However, errors in real world data have different origins and magnitude and may not follow specific probabilistic model. Sometimes, it is hard to know the true origin of corruptions present in data. Thus, to obtain method robust to arbitrary corruption we propose to introduce the CIM of the error. Rationale behind introduction of any regularization on $\mathbf{C}$ is to reflect its structural property of block-diagonality. Even though $\|\mathbf{C}\|_{S_p}$ and $\|\mathbf{C}\|_p$, $0 \leq p \leq 1$ in principle satisfy the enforced block-diagonality condition, their approximation of the BD structure of $\mathbf{C}$ is indirect (Lu et al., 2018). Hence, for comparison, this study proposes introduction of loss function with gradient-based BD regularization on representation matrix $\mathbf{C}$.

## 2.3 BASELINE MODEL DESCRIPTION

The base of the DSCNet (Ji et al., 2017) is a fully convolutional autoencoder. Following the flattened latent code of the autoencoder is an additional self-expression module which forms the full architecture of the DSCNet. While it is producing representation matrix $\mathbf{C}$, the clustering algorithm can only cluster the in-sample data. Thus, clustering error is not part of the learning process and, therefore, does not influence the quality of learned features and representation matrix. According to

the training procedure given in (Ji et al., 2019a), the performance of the DSCNet strongly depends on observing true labels every epoch. Although presented as an unsupervised method, DSCNet is using the minimum of clustering error as an early stopping criterion which is in contradiction with the principles of unsupervised learning and arguably leads to the overfitting and overly-optimistic performance estimation. As can be seen in Figure 2, much better representation matrix $\mathbf{C}$ can be learned from a model by ignoring loss and stopping when the accuracy is highest. Another problematic part is that the DSCNet algorithm is a post-processing matrix $\mathbf{C}$ with a hyper-parameter optimized using the ground truth labels. Post-processing of $\mathbf{C}$, as done in DSCNet (Ji et al., 2019a), has three steps. First, it is thresholded by keeping only $\delta$ largest, in terms of magnitude, elements in a row. The accepted number of largest elements depends on their sum - $S_\delta$. If the sum $S_\delta$ exceeds $\lambda \sum_{i=1}^{N} c_{ji}, \ i \in \{1, \ldots, N\}$, where $\lambda$ stands for an empirically set thresholding constant and $N$ represents the number of elements in a row, other elements (not included in the sum $S_\delta$) are set to zero. This first step is skipped in robust model. Second, knowing the dimensionality of the dataset $d$, after SVD decomposition of thresholded representation matrix $\mathbf{C}$ only $d$ largest eigenvalues with related eigenvectors were kept meaning the remaining eigenvectors span the noise subspace. Third, resulting values of Laplacian matrix are further suppressed by an exponentiation. The reconstructed representation matrix serves as an input to the spectral clustering algorithm which produces pseudo-labels.

Furthermore, DSCNet architecture is a base for the $S^2$ConvSCN model (Zhang et al., 2019a), which is why $S^2$ConvSCN suffers from the majority of problems as DSCNet.

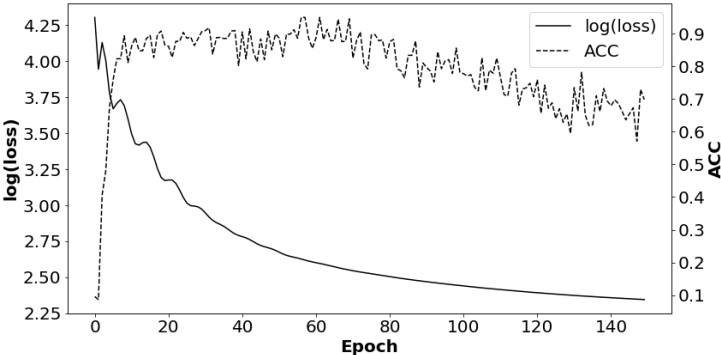

Figure 2: The figure illustrates the change of accuracy and loss over epochs for the original DSCNet model with MSE in the self-expressive layer. Model was trained and tested on a full COIL20 dataset with $\ell_2$ regularization of representation matrix $\mathbf{C}$ (see Equation (6)). It can be seen that accuracy is oscillating even though the loss is decreasing. Thus, selecting the model requires access to ground truth labels and makes an unfair practice.

A novelty that is presented in $S^2$ConvSCN is an FC layer, attached to the latent code, that is upgrading the DSCNet model. On its output, FC has softmax which can directly classify the input data. Upon finishing the learning procedure, the encoder and FC layer form a new model that can be used for evaluating the learning procedure on the independent unseen set. While authors of the $S^2$ConvSCN model (Zhang et al., 2019a) provide a tool for dealing with the out-of-sample data, the actual performance on unseen data has not been tested. The second novelty of $S^2$ConvSCN is training the FC layer from pseudo-labels. When training the network, the spectral clustering module clusters the data according to the affinity matrix every $E$ epochs and updates the pseudo-labels. It is important to note that the mentioned affinity matrix is constructed from the matrix $\mathbf{C}$ learned in the self-expression layer and it is changing every epoch during the training. Pseudo-labels assigned to the data are used in two ways. The first is to train the FC layer for soft classification, and the second is to suppress the $\mathbf{C}$ matrix values for samples that do not belong to the same cluster. Both ways are known as self-supervision. Keeping in mind that: (i) the $S^2$ConvSCN is upgraded DSCNet model, (ii) it is using the pretrained DSCNet parts of the network, and (iii) it has not been tested on the independent dataset, it is reasonable to conclude that the label-leakage also appeared in $S^2$ConvSCN. Regardless of having the self-supervision modules, the first pseudo-labels generated from DSCNet contain leaked knowledge about the group affiliations of the data.

Authors of both (Ji et al., 2017) and (Zhang et al., 2019a) suggest that to converge, parts of the model should be pretrained before training the whole model. However, there is no guarantee that the model will keep converging after attaching final layers or modules. For example, if a pretrained DSCNet yields reasonably good pseudo-labels, it is possible that after attaching the FC layer and performing self-supervision in S²ConvSCN, the matrix **C** will get worse and the whole model will diverge from the optimal solution. Thus, it is important to tune constants associated with the loss functions of the mentioned modules. As the S²ConvSCN has many loss functions regarding different layers or modules, a smaller learning rate could be beneficial in the sense that the overall loss can reach a better minimum in the error space. However, decreasing the learning rate on a plateau impacts the less-contributing losses. That can spoil the goodness of representation matrix **C** while the training process tries to reach the loss minimum.

## 3 ROBUST SELF-SUPERVISED CONVOLUTIONAL SUBSPACE CLUSTERING NETWORK

Motivated by discussion in previous section, we propose two new objective functions $\mathcal{L}_{CIM}$ for the self-expression module of the S²ConvSCN architecture:

$$\min_{\mathbf{C}} \mathrm{CIM}^2(\mathbf{E}) + \gamma \left\| \mathbf{C} \right\|_{[k]},\tag{5}$$

$$\min_{\mathbf{C}} \mathrm{CIM}^2(\mathbf{E}) + \gamma \left\| \mathbf{C} \right\|_2.\tag{6}$$

where **E** is defined in Eq. (3). $\left\| \mathbf{C} \right\|_{[k]}$ denotes BD regularization and $\left\| \mathbf{C} \right\|_2$ denotes $\ell_2$ regularization of representation matrix **C**. $\gamma$ represents a trade-off constant. The CIM loss is defined in Appendix A.2. Objectives (5) and (6) ensure a smooth decrease of the loss function that enables the use of label-free stopping criterion. For the sake of completeness, other loss functions of Robust S²ConvSCN are stated. Auto-encoder reconstruction loss, where **X** represents the input data and $\hat{\mathbf{X}}$ represents the output of the decoder is defined as:

$$\mathcal{L}_{REC} = \frac{1}{2N} \sum_{j=1}^{N} \left\| \mathbf{x}_j - \hat{\mathbf{x}}_j \right\|_2^2 = \frac{1}{2N} \left\| \mathbf{X} - \hat{\mathbf{X}} \right\|_F^2.\tag{7}$$

Pseudo-labels form a matrix **Q** where element $q_{ij}$ is set to 1 if samples $i$ and $j$ have the same pseudo-label. Otherwise, $q_{ij}$ is set to 0. When **Q** is calculated, $\left\| \mathbf{C} \right\|_{\mathbf{Q}}$ loss is defined as:

$$\mathcal{L}_{CQ} = \sum_{i,j} |c_{ij}| \frac{\left\| \mathbf{q}_i - \mathbf{q}_j \right\|_2^2}{2} := \left\| \mathbf{C} \right\|_{\mathbf{Q}}\tag{8}$$

where $\mathbf{q}_i$ and $\mathbf{q}_j$ represent one-hot encoded pseudo-labels for $i$-th and $j$-th sample. $|c_{ij}|$ stands for the absolute representation value for the corresponding samples in the representation matrix **C**. Cross-entropy and center loss of FC layer is:

$$\mathcal{L}_{CE} = \frac{1}{N} \sum_{j=1}^{N} ln(1 + e^{\hat{\mathbf{y}}_j^T \mathbf{q}_j}), \ \mathcal{L}_{CNT} = \frac{1}{N} \sum_{j=1}^{N} \left\| \mathbf{y}_j - \mu_{\pi(\mathbf{y}_j)} \right\|_2^2\tag{9}$$

where $\hat{\mathbf{y}}_j^T$ stands for a softmax normalized output and $\mathbf{y}_j$ represents the logits of the FC layer for $j$-th sample. $\mu$ represents a centroid of a cluster $\pi$ taken from the spectral clustering output for a given sample $j$ (Zhang et al., 2019a). Additionally, in order to yield a stable and unique solution, representation matrix **C** is forced to be symmetrical (Lu et al., 2018). A novelty introduced in this robust version of the network is a symmetric loss which is defined as:

$$\mathcal{L}_{SYM} = \frac{1}{2} \sum_{j=1}^{N} \left\| \mathbf{c}_j - \mathbf{a}_j \right\|_2^2 = \frac{1}{2} \left\| \mathbf{C} - \mathbf{A} \right\|^2\tag{10}$$

where $\mathbf{A}$ represents symmetric affinity matrix. Every loss function has an assigned trade-off regularization constant $\lambda_1$ to $\lambda_5$ to regulate its importance. Thus, the total loss function, with novelties bolded, is defined as follows:

$$\mathcal{L}_T = \mathcal{L}_{REC} + \lambda_1 \mathcal{L}_{\mathbf{CIM}} + \lambda_2 \mathcal{L}_{CQ} + \lambda_3 \mathcal{L}_{CE} + \lambda_4 \mathcal{L}_{CNT} + \lambda_5 \mathcal{L}_{\mathbf{SYM}}. \tag{11}$$

Furthermore, to address the overfitting issue, we present results both on the train sets and on the independent test sets. The latter will give better insights into the capability of used algorithms. Also, $\mathbf{C}$ matrix post-processing is reduced by skipping the first out of three steps described in Subsection 2.3. A more informative illustration of the complete architecture (shared by the baseline $S^2$ConvSCN and robust model) is shown in Figure 3.

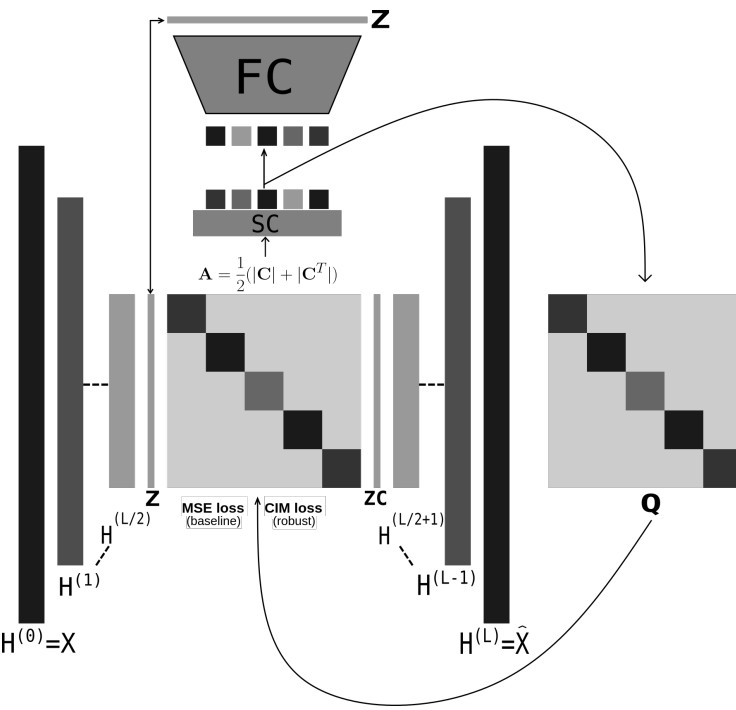

Figure 3: Self-supervised robust (and baseline) $S^2$ConvSCN architecture showing all layers and modules. Encoder and decoder layers are denoted by $H^{(0)}$ to $H^{(L)}$. Together, flattened encoder features $\mathbf{Z}$ and representation matrix weights $\mathbf{C}$ construct self-expression layer $\mathbf{ZC}$ where optimal $\mathbf{C}$ is learned. Spectral clustering (SC) module receives affinity matrix $\mathbf{A}$ as an input. Pseudo-labels from SC module serve as target labels for the FC-softmax classifier. The matrix $\mathbf{Q}$ is used as a regularizer for the representation matrix $\mathbf{C}$ to learn a better representation.

## 4 EXPERIMENTAL RESULTS

In this section, we compare the clustering performance of Robust $S^2$ConvSCN with state-of-the-art self-supervised $S^2$ConvSCN model on four well-known datasets. Because such comparison would be unfair to the Robust $S^2$ConvSCN model, performance is not compared to the performances of shallow and recent deep subspace clustering methods reported in the literature. Shallow models use labels to optimize hyperparameters and then evaluate the performance based on these labels. Our approach used only the in-sample labels for hyperparameter-tuning, while out-of-sample labels were kept for evaluation. A worth-noting alternative approach of hyperparameter optimization that does not require labels was presented in (Lipor & Balzano, 2020). However, their approach is not based on raw data but rather on extracting features using scattering network. Regarding the label-leakage issue from pretrained parts of the network (see Figure 2), this study avoids this problem by learning from scratch within defined standards. The performance is evaluated in term of accuracy (Acc):

$$Acc(\hat{\mathbf{r}}, \mathbf{r}) = \max_{\pi \in \Pi_k} \left( \frac{1}{N} \sum_{i=1}^{N} \{\pi(\hat{r_i}) = \mathbf{r}_i\} \right) \tag{12}$$

where $\Pi_k$ stands for the permutation space of $[k]$ defined as all possible orderings of the $k$-element set $\{1, 2, \ldots, k\}$. We compare the performance of Robust S$^2$ConvSCN with state-of-the-art self-supervised deep subspace clustering algorithm (Zhang et al., 2019a). ADAM optimizer (Kingma & Ba, 2014), an extension to stochastic gradient descent is used in the proposed learning procedure. For a full implementation details, see Appendix B. Hyperparameter settings and regularization constants are shown in Tables 3 and 4 in Appendix B.

### 4.1 COMPARISON ON FULL DATASETS

As previously discussed, learning and evaluating on the same data yields an overly optimistic estimation of the model's performance. Due to the fair comparison, we implemented and compared our model to the different versions of the S$^2$ConvSCN model in the same manner as reported in (Zhang et al., 2019a). However, we did not use accuracy as a stopping criterion. Instead, a fixed number of epochs, learning rate decay, and early stopping based on loss decrease only were used. Same settings are used for the baseline and robust version (see Table 3 in Appendix B). Thus, the obtained results differ significantly from (Zhang et al., 2019a) report. For S$^2$ConvSCN, optimal regularization constants were transferred from (Zhang et al., 2019a). In Table 1, a comparison of S$^2$ConvSCN and Robust S$^2$ConvSCN with different $\mathbf{C}$ matrix regularization strategies can be seen. When comparing Figure 2 and Figure 4, it can be seen that CIM prevents overtraining of the network and that loss can be used for early stopping criterion.

Table 1: Accuracy comparison of S$^2$ConvSCN and Robust S$^2$ConvSCN trained and tested on full COIL20, COIL100, and Extended YaleB datasets. BD and L2 stand for block diagonal and $\ell_2$ regularization of representation matrix, respectively.

|  | COIL20 | COIL100 | EYaleB |
|---|---|---|---|
| **S$^2$ConvSCN + BD** | 0.81111 | 0.30375 | 0.56867 |
| **S$^2$ConvSCN + L2** (Zhang et al., 2019a) | 0.63333 | 0.55319 | 0.75247 |
| **RS$^2$ConvSCN + BD** | 0.76250 | 0.50528 | 0.41159 |
| **RS$^2$ConvSCN + L2** | **0.88403** | **0.68805** | **0.81661** |

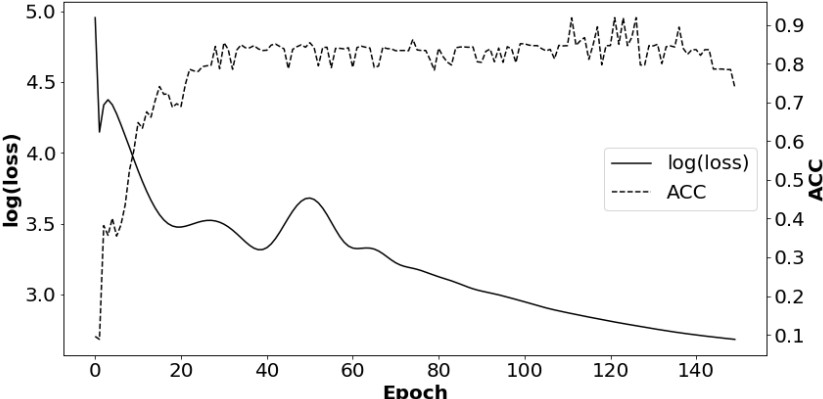

Figure 4: The figure illustrates the change of accuracy and loss over epochs for a Robust DSCNet model with CIM in the self-expressive model. In comparison with Figure 2, it can be seen that Robust DSCNet can use the change in loss as CIM prevents overfitting.

## 4.2 Classification performance on unseen data

Having several independent observations is crucial for the model's performance validation. Thus, according to Table 3 in Appendix B, datasets were split to stratified folds. For MNIST, as there is enough data, each fold was additionally divided into a train and test set, 70% and 30%, respectively. For other datasets, the remaining folds were all used for testing as these datasets have a very limited number of data per label, i.e. additional splitting as for MNIST could not be possible. In the context of (Li et al., 2020), the obtained results in Table 2 support the claim that the network is robust to noise when early-stopping methods are applied.

Table 2: Accuracy comparison of $S^2$ConvSCN and Robust $S^2$ConvSCN trained and tested on independent folds for each dataset. BD and L2 stand for block diagonal and $\ell_2$ regularization of representation matrix, respectively. The best mean accuracy for each dataset is bolded.

| | | MNIST | COIL20 | COIL100 | EYaleB |
|---|---|---|---|---|---|
| **$S^2$ConvSCN + BD** | **mean** | 0.18721 | 0.71686 | 0.33762 | 0.49187 |
| | **stddev** | 0.06258 | 0.07814 | 0.04736 | 0.04261 |
| **$S^2$ConvSCN + L2** (Zhang et al., 2019a) | **mean** | 0.12659 | 0.67486 | 0.52099 | 0.63735 |
| | **stddev** | 0.02569 | 0.05917 | 0.02295 | 0.07062 |
| **RS$^2$ConvSCN + BD** | **mean** | **0.50417** | 0.79824 | 0.44882 | 0.50017 |
| | **stddev** | 0.05329 | 0.02984 | 0.03845 | 0.02695 |
| **RS$^2$ConvSCN + L2** | **mean** | 0.42979 | **0.82547** | **0.55461** | **0.75239** |
| | **stddev** | 0.04878 | 0.02537 | 0.01454 | 0.03346 |

## 5 Discussion and conclusion

As it could be seen in Table 2, Robust $S^2$ConvSCN outperforms $S^2$ConvSCN on all datasets regardless of regularization imposed on representation matrix **C**. Moreover, in the case of the MNIST dataset, Robust $S^2$ConvSCN with BD regularization leads to the best results. The CIM loss, indeed, handles the corruptions in the data better than MSE, especially the possible amplification of corruptions due to their propagation through deep layers. The noise robustness can be explained by Chen et al. (2016) where is shown that learning using the correntropy-loss function generalizes better. However, training of deep learning algorithms is an especially time-consuming task. Thus, settings in Appendix Table 3 and regularization constants could possibly be improved in further testing. As discussed in earlier sections, we assume that (Zhang et al., 2019a) approach leads to an overly optimistic estimation of the model's performance. Nevertheless, Robust $S^2$ConvSCN outperformed $S^2$ConvSCN showing its superiority.

Using datasets for which the dimensionality $d$ is unknown could be a challenging task. Without that information, it is difficult to post-process the representation matrix to eliminate the noise (Haeffele et al., 2020). Also, there is a problem with pseudo-labels refinement frequency. If it occurs too often it could lead to divergence of the whole model. If the refinement occurs too rarely, the model could get stuck in the poor minimum. Finding a more robust pseudo-label refinement strategy and elimination of the need for the prior knowledge of $d$ is still an open question. Also, more research towards identifying which loss has higher significance during the training is needed as there are many non-equally important loss functions in Robust $S^2$ConvSCN. In a sense of memory constraints, mini-batch training of deep subspace clustering models could cross the barrier of memory complexity and offer more approachable learning.

To conclude, among finding a more robust model with better early stopping criterion, this ablation study aimed to set up a new, more transparent way for the evaluation of deep subspace clustering models. As baseline algorithms experience early commitment problem, i.e. they depend on weight-transfer from pretrained models, true labels are leaked during the training process. For that, we propose an evaluation of the algorithm on independent data to have a proper estimate of model performance. As can be seen from Table 2, measured performances of $S^2$ConvSCN significantly differ from the optimistic one presented in (Zhang et al., 2019a). The combination of gradient-based learning with CIM loss and early-stopping strategy (Chen et al., 2016; Li et al., 2020) did, indeed, improve the robustness to the unknown errors in data. Additionally, the presented model which incorporates label-free learning and the robust correntropy loss can easily be extended to multi-modal and multi-view data. We aim to address that in our further research.

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

## A    APPENDIX - PRELIMINARIES

### A.1    BLOCK DIAGONAL REGULARIZATION

To introduce BD regularization we state the proposition 4 from (Von Luxburg, 2007) for the graph Laplacian matrix $\mathbf{L}$:

**Proposition 1.** (Proposition 4 in (Von Luxburg, 2007): Number of connected components and spectra of $\mathbf{L}$). Let $\mathbf{G}$ be an undirected graph with nonnegative weights. Then the multiplicity $k$ of the eigenvalues 0 of $\mathbf{L}$ equals the number of connected components in the graph. Thus, based on Proposition 1 we define the BD regularization of $\mathbf{C}$ as the sum of the $k$ smallest eigenvalues of $\mathbf{L}$:

$$\|\mathbf{C}\|_{[k]} = \sum_{i=N-k+1}^{N} \lambda_i(\mathbf{L}) \tag{13}$$

where $\lambda_i(\mathbf{L})$, $\lambda_1 \geq \lambda_2 \geq \cdots \geq \lambda_{(N-k+1)} \geq \cdots \geq \lambda_N$, stands for the $i$-th eigenvalue of $\mathbf{L}$.

## A.2 CORRENTROPY

Here we briefly introduce the correntropy and its properties that qualify it as loss function robust to data corruptions. Let $\mathbf{S} = [\mathbf{s}_1, \ldots, \mathbf{s}_N] \in \mathbb{R}^{d \times N}$ and $\mathbf{T} = [\mathbf{t}_1, \ldots, \mathbf{t}_N] \in \mathbb{R}^{d \times N}$ be two realizations of the corresponding random variables. The empirical correntropy is estimated from data as:

$$\hat{V}(\mathbf{S}, \mathbf{T}) = \frac{1}{N} \sum_{i=1}^{N} \kappa_\sigma(\mathbf{s}_i, \mathbf{t}_i), \ \kappa_\sigma(\mathbf{s}_i, \mathbf{t}_i) = \exp\left(-\frac{\|\mathbf{s}_i - \mathbf{t}_i\|}{2\sigma^2}\right) \tag{14}$$

where $\kappa_\sigma(\mathbf{s}_i, \mathbf{t}_i)$ is the Gaussian kernel. Herein, we present from (Liu et al., 2007) two (out of ten) properties of correntropy that justify its use as robust error measure.

**Property 1.** (Property 3 in (Liu et al., 2007)). Correntropy involves all the even moments of the random variable $\varepsilon = \mathbf{T} - \mathbf{S}$:

$$V_\sigma(\mathbf{S}, \mathbf{T}) = \frac{1}{\sqrt{2\pi}\sigma} \sum_{n=0}^{\infty} \frac{(-1)^n}{2^n n!} \mathbb{E}\left[\frac{(\mathbf{S} - \mathbf{T})^{2n}}{\sigma^{2n}}\right] \tag{15}$$

where $\mathbb{E}$ denotes mathematical expectation. As $\sigma$ increases, the high-order moments decay faster; so, the second-order moment tends to dominate, and the correntropy approaches correlation.

While the MSE involves only second-order moments and is, thus, optimal measure for error distributed normally, correntropy is the optimal measure for error with the arbitrary (non-Gaussian) distribution. Furthermore, as can be seen from Equation (14), correntropy is data-driven.

**Property 2** (Property 8 in (Liu et al., 2007)). The function:

$$CIM(\mathbf{S}, \mathbf{T}) = \left(\kappa(\mathbf{0}, \mathbf{0}) - \hat{V}(\mathbf{S}, \mathbf{T})\right)^{\frac{1}{2}} \tag{16}$$

defines a CIM in sample space. Since its limes is final, the CIM function is robust to large random errors.

## B APPENDIX - EXPERIMENTAL SETUP

Robust $S^2$ConvSCN model is implemented in Keras (Chollet et al., 2015) and Tensorflow (Abadi et al., 2015). First, the random seed is fixed for reproducibility. To have a reliable performance estimate on the test set and multiple independent train and test folds for each dataset, a stratified k-fold splitting was performed. Depending on the dataset constraints, the train-test split was performed on every fold or one fold served as a train set while the rest $k - 1$ folds served as a test set. Hence, independent $k$ observations of algorithms' performances for each dataset have been produced.

As discussed in Section 3, instead of monitoring accuracy every epoch, and consequently supervising the learning using ground truth labels, we use only the loss for reducing learning rate and early

stopping. The training is stopped either after reaching the early stopping criterion or after reaching the maximum number of epochs. As in (Zhang et al., 2019a), pretraining was also applied. Firstly, the autoencoder is pretrained to replicate the input. Secondly, the self-expression layer together with the pretrained autoencoder (DSCNet (Ji et al., 2017)) was trained to reach a limited number of epochs or the early stopping criterion. After that, the FC layer and self-supervision modules were added to the pretrained DSCNet. This procedure forms a starting point for both $S^2$ConvSCN and Robust $S^2$ConvSCN. Values of hyper-parameters $\lambda_1 \ldots \lambda_5$ found in Eq. 11 are shown in Table 4.

During the warming-up phase, pseudo-labels were not refined because the newly introduced layers and modules could affect the convergence process (see Section 3 for the discussion). For the sake of fair comparison, different versions of $S^2$ConvSCN were trained using the same settings as Robust $S^2$ConvSCN. Table 3 shows settings for MINST (LeCun et al., 2010), COIL-20 (Nene et al., 1996b), COIL-100 (Nene et al., 1996a), and Extended Yale B (Lee et al., 2005). Detailed settings for the baseline and robust model can be seen in Table 3.

Table 3: Settings of the baseline $S^2$ConvSCN and Robust $S^2$ConvSCN per each dataset. T-max represents maximum number epochs and T0 stands for the number of epochs after which pseudo-labels are refined. Warm-up represents the number of epochs without the refinement of pseudo-labels, LR stands for starting learning rate, and Min LR represents the minimum learning rate.

|  | MNIST | COIL20 | COIL100 | EYaleB |
|---|---|---|---|---|
| **data splits** | 15 | 5 | 5 | 5 |
| **T-max** | 9000 | 4000 | 4000 | 9000 |
| **T0** | 30 | 50 | 40 | 30 |
| **Warm-up** | 50 | 100 | 80 | 50 |
| **LR** | 1e-3 | 1e-4 | 1e-4 | 1e-4 |
| **Min LR** | 1e-6 | 1e-6 | 1e-7 | 1e-6 |

Table 4: $S^2$ConvSCN and RS$^2$ConvSCN hyperparameters. $\lambda_1$ to $\lambda_5$ are regularization constants from total loss Eq. 11 while the trade-off constant $\gamma$ depends on usage of BD (Eq. (5)) or $\ell_2$ (Eq. (6)) regularization.

|  | Model | $\lambda_1$ | $\lambda_2$ | $\lambda_3$ | $\lambda_4$ | $\lambda_5$ | $\gamma$ Eq. (5) | $\gamma$ Eq. (6) |
|---|---|---|---|---|---|---|---|---|
| **Coil20** | **S$^2$ConvSCN** | 30 | 8 | 6 | 3 | 1 | 3e-2 | 3e-2 |
|  | **RS$^2$ConvSCN** | 300 | 8 | 6 | 3 | 1 | 3e-3 | 3e-3 |
| **Coil100** | **S$^2$ConvSCN** | 30 | 8 | 7 | 3.5 | 1 | 3e-2 | 3e-2 |
|  | **RS$^2$ConvSCN** | 600 | 8 | 7 | 3.5 | 1 | 1e-3 | 1e-3 |
| **EYaleB** | **S$^2$ConvSCN** | 6.3 | 1 | 72 | 36 | 2 | 3e-2 | 3e-2 |
|  | **RS$^2$ConvSCN** | 6.3 | 1 | 72 | 36 | 2 | 3e-2 | 3e-2 |
| **MNIST** | **S$^2$ConvSCN** | 30 | 1 | 72 | 36 | 1 | 3e-2 | 3e-2 |
|  | **RS$^2$ConvSCN** | 30 | 1 | 72 | 36 | 1 | 3e-2 | 3e-2 |

## C APPENDIX - COMPLEXITY AND CONVERGENCE

Dataset size and choice of representation matrix regularization have a direct impact on time and memory complexity. As the model must learn in batch mode, the number of samples in the dataset $N$ determines the number of parameters in the self-expression layer - $N^2$. Thus, dataset size impacts memory and time complexity. Also, regularization imposed on the representation matrix affects the computational time of the algorithm. As presented in Section A, BD regularization is defined as the sum of $k$ smallest eigenvalues of Laplacian matrix $\mathbf{L}$. Thus, if the BD regularization is chosen, the model will have higher time complexity due to the usage of the singular value decomposition algorithm. Although it is not directly associated with the time complexity per epoch, the spectral clustering module refines pseudo-labels every $T0$ epochs which adds to overall training time.

To properly emphasize individual loss functions, all regularization constants in the overall loss function must be tuned (discussed in Section 3). As a good starting point, we used regularization constants from (Zhang et al., 2019a) and refined only those associated with the error measure term in

the self-expression layer and representation matrix $\mathbf{C}$. Depending on the fold size and regularization used, training of one fold on one Nvidia Quadro P6000 GPU takes 2-6 hours. Testing on one fold is within several seconds.

