# OpenReview forum: "Subspace Clustering via Robust Self-Supervised Convolutional Neural Network"
_ICLR.cc/2021/Conference — Reject_

### Official Review · AnonReviewer3 · 2020-10-28
**Need a better presentation style and clarification**

**Rating:** 5
**Confidence:** 4

**Review:**

The authors propose to add a correntropy induced metric (CIM) loss term to improve the robustness of the self-supervised convolutional subspace clustering network (${\rm S}^2$ConvSCN) to data corruption. The authors show that in a truly unsupervised training environment, the proposed robust ${\rm S}^2$ConvSCN method outperforms the original ${\rm S}^2$ConvSCN in [1].

Pros:
1. The authors point out a very interesting fact that many deep subspace clustering methods often require access to the ground-truth labels to decide the stopping of the learning process and optimize hyperparameters using external clustering validation methodology, whereas the same labels are used for tuning hyperparameters and evaluating the final clustering performance.

Cons:
1. The presentation style of this paper needs improvement. For example:
    1.1 As this paper is based on the comparison of [1], the differences in Figure 1 should be clearly stated. Currently, this comparison is mixed in the contribution part, which is really confusing. The reviewer would recommend that before showing the results, the authors elaborator the differences between the proposed method and the method in [1] point by point. Then clearly state which part of the method in [1] is modified for "fair" comparison. This is very important for evaluating the contributions of this paper.
    1.2 Define $\bf{E}$ in (1).
    1.3 Please define CIM near (4) and (5), or at least explicitly give the reference to your Appendix. Same for $||\bf{C}||_{[k]}$. The reviewer believes that this paper is modified from [2], but the presentation flow is not carefully addressed. The Arxiv version is more readable.
    1.4 Define $\bf{Q}$ shown between (6) and (7).
    1.5 The reviewer recommends that the authors include their model structure.

2. The authors mention "unfair" comparison in the paper. I think a possible fair way to compare the performance of ${\rm S}^2$ConvSCN method and robust ${\rm S}^2$ConvSCN method on an "unseen" dataset.

3. Please give your hyper-parameters in the paper.

4. Please define the permutation space of $[k]$ in (11).

5. For "fair" comparison, the author modified the method in [1]. However, in 4.1, only "fixed number of epochs, learning rate, decay, and early stopping based on loss decrease only were used". These should be explicitly stated. Are they the same as those in Table 3?

6. The authors also mention the training time in the paper. However, for the device, only the word "GPU" is mentioned in the last paragraph without any details.

In summary, I believe this paper can be interesting if the presenting issue is well-addressed, and the comparison is well-stated, but the current version is not really good for acceptance.


Reference:
[`1] Junjian Zhang, Chun-Guang Li, Chong You, Xianbiao Qi, Honggang Zhang, Jun Guo, and Zhouchen Lin. Self-supervised convolutional subspace clustering network. In Proceedings of the IEEE Conference on Computer Vision and Pattern Recognition, pp. 5473-5482, 2019a.
[2] https://arxiv.org/abs/2004.03375

----
After rebuttal:

Thank the authors for providing a revised version. The paper is more readable, however, I still hesitate to give 6 for the current version.  I would say that the fact pointed in this paper is really interesting, but this paper may need further improvement in the presentation.

---

> ### Author Response · Authors · 2020-11-15
> **Answers to AnonReviewer3 concerns**
>
> *The authors propose to add a correntropy induced metric (CIM) loss term to improve the robustness of the self-supervised convolutional subspace clustering network (ConvSCN) to data corruption. The authors show that in a truly unsupervised training environment, the proposed robust ConvSCN method outperforms the original ConvSCN in [1].
> Pros:*
> 1) *The authors point out a very interesting fact that many deep subspace clustering methods often require access to the ground-truth labels to decide the stopping of the learning process and optimize hyperparameters using external clustering validation methodology, whereas the same labels are used for tuning hyperparameters and evaluating the final clustering performance.*
> - We thank the reviewer for this assessment. We are free to add that besides mentioned fact, we proposed modification of the existing S2ConvSCN by replacing the MSE-loss term with the CIM-loss term. That is demonstrated to enable the robust S2ConvSCN to learn in the truly label-free manner. We also found that combination of gradient-based learning, with the CIM loss and early-stopping strategy, improved robustness to the presence of the unknown errors in data. In particular, the CIM loss yields smooth learning which enables loss-based label-free early stopping.
>
> *Cons:*
> 1) *1st comment*
> -  We thank the reviewer for this comment. We will revise the paper taking into account all the reviewers' comments the best we can. In the revised paper we stated in Figure 1 the key differences between the baseline and   proposed robust version of the network. We modified Figure 1, including the caption, accordingly.
> 1) *Comment 1.1*
> - Furthermore, at the end of Introduction we stated the key contributions of the paper in an itemized manner. The emphasized key contributions are: 1) CIM-loss substituted the MSE-loss. Robustness of the CIM loss to data corruptions of arbitrary origin led to the smoother training process. 2) thanks to 1), the network is trained using the early stopping by monitoring the accumulated loss only (no access to the labels is required). 3) combination of the CIM-loss and early stopping the training is less sensitive to data corruptions including the label noise which enables the trained network to generalize better. 4) the performance of proposed robust S2ConvSCN is evaluated on the unseen (out-of-sample) data. Thanks to 3), results presented in Table 2, outperformed the ones obtained by the baseline S2ConvSCN. 5) as a minor contribution we integrated block-diagonal regularization of self-representation matrix into gradient descent learning process. 6) post-processing of self-representation matrix is reduced to a significant extent.
>
> 1) *Comment 1.2*
> - We thank the reviewer for this comment. This is a mistake. **E** was supposed to stand for the modelling error (noise). However, since the self-expression model is exact there should be no error in Eq. (1). The error **E** appears in Eq.(2) when noise is introduced in the model. We fixed this problem in revised paper and defined **E** in Eq. (3).
>
> 1) *Comment 1.3*
> - We thank the reviewer for this comment. The reference to the Appendix A.2. is added to the revised paper. The ArXiv version was more readable because of the space limitation for the ICLR paper.
> 1) *Comment 1.4*
> - We thank the reviewer for this comment. In the revised paper, we defined the matrix Q.
> 1) *Comment 1.5*
> - We thank the reviewer for this comment. In the revised paper we concluded the Section 3 by stating the key differences (contributions) relative to the baseline network and illustrated the structure of the Robust S$^2$ConvSCN in newly introduced Figure 3.
>
> 2) *2nd comment*
> - We thank the reviewer for this comment. She/he is right. The fair way is to compare the performance of both versions on "unseen" data. That, actually, is presented in Table 2. However, what makes the comparison "unfair" is that the baseline network and majority of previous published models have to have access to the labels during the training process. Nevertheless, despite this "advantage", the baseline version performed worse on both "seen" and "unseen" data, see Tables 1 and 2.
>
> 3) *3rd comment*
> - We thank the reviewer for this comment. In revised paper we gave the values of the     hyper-parameters in Table 4 (Appendix B) and referenced the table in Section 4.
> 4) *4th comment*
> - We thank the reviewer for this comment. In the revised version of the paper we defined the permutation space of [k] as all possible orderings of the k-element set {1, 2, ..., k}.
> 5) *5th comment*
> - We thank the reviewer for this comment. Yes, the parameters are the same as in Table 3 for the Robust S2ConvSCN. In the revised paper, we stated that more clearly in Subsection 4.1.
> 6) *6th comment*
> - We thank the reviewer for this comment. Indeed, we missed to state that the type of the GPU card is NVIDIA QUADRO P6000. This is fixed in the revised version of the paper.

---

### Official Review · AnonReviewer2 · 2020-10-28
**A deep subspace clustering approach that aims to overcome some issues with existing methods but does not present compelling results.**

**Rating:** 3
**Confidence:** 4

**Review:**

This paper presents an approach to deep subspace clustering based on minimizing the correntropy induced metric (CIM), with the goal of establishing when training should be stopped and generalizing to unseen data. The main contribution over the existing S2ConfSCN method is a change from squared error loss to CIM when optimizing over the affinity matrix. A key benefit of CIM as a loss is that it does not decrease arbitrarily with training epochs, so it provides a means of estimating when training should cease without needing ground truth labels. The authors argue that CIM "ensures a smooth decrease of the loss function that enables the use of label-free stopping criterion." However, this claim is only justified through a minimal empirical evaluation. The authors also include a means of enforcing block diagonal structure in the learned affinity matrix.

While the ideas presented in the paper are important for the area of deep subspace clustering, the overall contribution of this paper is quite limited. Further, the results in Table 1 do not improve on the results of shallow subspace clustering methods such as SSC and EnSC. The authors argue that these methods are tuned using ground truth labels, but even tuning with the KSS Cost (as in Lipor and Balzano 2020) results in better performance on the given datasets. Given that shallow subspace clustering methods maintain strong theoretical guarantees, the empirical results presented here do not justify the use of deep methods.

---

> ### Author Response · Authors · 2020-11-15
> **Answers to AnonReviewer2 concerns**
>
> *First paragraph of the review*
>
> - We thank the reviewer for this comment. We argued that CIM "ensures a smooth decrease of the loss function that enables the use of label-free stopping criterion" based on the fact that the CIM loss, unlike the MSE loss, is robust to data corruptions of the arbitrary (unknown) origin. That is a consequence of the Property 1 (Eq. 15 in Appendix A) on the correntropy that, as opposed to the MSE, involves all the even moments of the error random variable. We believe that this mathematically proven property of the correntropy in combination with the empirical evaluation on four datasets suffices to support the stated claim.
>
> *Second paragraph of the review*
>
> - We thank the reviewer for this comment. The motivation for the present paper was to modify  the existing self-supervised network to learn in the truly label-free manner. It is found that combination of gradient-based learning with the CIM loss and early-stopping strategy improved robustness to the presence of the unknown errors in data. In particular, the CIM loss yields smooth learning, which enables loss-based label-free early stopping. In this regard, the ablation studies were conducted relative to the original (MSE-loss based) version of the network, with the results presented in Tables 1 and 2, to demonstrate the outlined finding. We stated in the Discussion and Conclusion section that “more research towards identifying which loss has higher significance during the training is needed as there are many non-equally important loss functions in Robust S2ConvSCN”. Hence, we were aware that even better performance that the one reported in Tables 1 and 2 can be expected. However, as we also explained, training of one fold on one GPU takes 2-6 hours. Therefore, grid-search validation of the hyper-parameters, in the way performed with the KSS Cost as in Lipor and Balzano 2020, requires an immense amount of time.
>
> - Also, we would like to quote the mentioned paper (Lipor and Balzano 2020): *“We evaluate the proposed CQMs on synthetic data as well as three ofthe most common benchmark datasets in the subspace clustering literature: the Hopkins-155 dataset [54], the cropped Extended Yale Face Database B [6, 55], and the COIL-20 [56] object database, **with preprocessing steps performed as in [23]**.”* Preprocessing steps from the mentioned reference [23] (Lipor et.al., Subspace Clustering using Ensembles of K-Subspaces) include feature extraction from scattering network: *“We consider the Hopkins-155 dataset [2], the cropped Extended Yale Face Database B [4,45],COIL-20 [46] and COIL-100 [47] object databases, the USPS dataset provided by [48], and 10,000 digits of the MNIST handwritten digit database [5], **where we obtain features using a scattering network** [49] as in [18]”.*
>
> - While not underestimating the importance of their work, we argue that the performance achieved in (Lipor and Balzano 2020) heavily depends on selection of the feature extraction technique. However, the input to our model was raw data and the network has to cluster the data and extract the best possible features, simultaneously. That stands for a harder task than just preforming clustering.  However, in our work, we were not trying to achieve the best possible performance on the benchmark datasets. Our intent was to show the limitations of previous deep subspace clustering techniques and guide the further research to better practices. We stress that our empirical results, thus, show the possible way to overcome the problem of tuning deep models on ground truth labels by introducing the CIM loss.
>
> *Given that shallow subspace clustering methods maintain strong theoretical guarantees, the empirical results presented here do not justify the use of deep methods.*
> - While we agree that shallow subspace clustering models have strong theoretical guarantees, we argue that deep methods should be separately developed further because of their potential. As in other fields (e.g. computer vision), certain good practices have to be established in order to, eventually, yield empirically (and possibly theoretically) justified applications. This is what we aimed with our paper. It is our intention in the forthcoming work to perform further validation of the loss terms in the composite loss function, as well as to further reduce (eliminate) the post-processing of the representation matrix. That is expected to yield the Robust S2ConvSCN truly independent from the prior knowledge. In addition to detailed ablation study, we also do plan to perform a more comprehensive performance analysis.

---

### Official Review · AnonReviewer1 · 2020-10-29
**Paper537 review**

**Rating:** 5
**Confidence:** 5

**Review:**

This paper proposes the robust formulation of the self-supervised convolutional subspace clustering network. In order to handle the data corruptions, the correntropy induced metric (CIM) of the error is embedded into the loss function. I have some concerns about this paper:
1. The major issue of this work is the limited novrlty. In its current form, the only novel point is the CIM loss of the representation error of Z in Eq.(4) and (5). Other loss terms in Eq.(10) are all used in previous works.
2. There are too many parameters in Eq.(10). However, no ablation analysis for different loss terms are given.
3. Experimental comparison is not sufficient. Only one method is used for comparison. However, there are many deep learning based clustering methods, they should be used for comparison.

---

> ### Author Response · Authors · 2020-11-15
> **Answers to AnonReviewer1 concerns**
>
> *This paper proposes the robust formulation of the self-supervised convolutional subspace clustering network. In order to handle the data corruptions, the correntropy induced metric (CIM) of the error is embedded into the loss function. I have some concerns about this paper:*
> 1) *The major issue of this work is the limited novelty. In its current form, the only novel point is the CIM loss of the representation error of Z in Eq.(4) and (5). Other loss terms in Eq.(10) are all used in previous works.*
> - We thank the reviewer for this comment. As pointed out in the paper, our motivation was to develop a version of the self-supervised deep network capable of truly label-free (unsupervised) learning of the weights. We found that combination of gradient-based learning with the CIM loss and early stopping strategy improve robustness to the presence of the unknown errors in data. Apart from introducing the CIM loss, this research aimed to propose a way to overcome the problem of early stopping. As discussed in the paper, majority of deep subspace clustering methods tune their parameters using an unfair practice: early stopping of the training procedure based on the ground-truth labels which is followed by the evaluation on the same set of labels. The primary reason it was done in previous research was the MSE based loss and its problem of leading a model to overfit. In particular, the CIM loss yields smooth learning, which enables loss-based label-free early stopping. The primary intent of this research was, however, to set up a more fair practice in deep subspace clustering research, introducing at the same time the CIM loss as a practical solution to the problem of early stopping.
> - Regarding Eq. (10), the last term related to symmetric loss function is also a novelty introduced in modified robust version of the network. However, other parts of the composite loss function are of less importance in comparison to the CIM loss. Their relative importance is planned to be evaluated in the future work. All novelties, including regularizations, early-stopping technique, and evaluation procedure are illustrated in Figure 1 and listed in the introduction section of the revised paper.
> 2) *There are too many parameters in Eq.(10). However, no ablation analysis for different loss terms are given.*
> - We thank the reviewer for this comment. Majority of loss terms (REC, CQ, CE, CNT) in Eq.(10) were introduced in previous research and, together with MSE and L-2 regularization of self-expression matrix, make a baseline model. As explained previously, we were merely focused to upgrade the existing self-supervised network to learn in the truly label-free manner. We understood that the major reason that prevented label-free learning of the existing version of the network was the oscillatory nature of the learning process caused by the sensitivity of the MSE loss function to non-Gaussian corruptions present in data. Thus, label-based accuracy criterion was necessary to stop the learning process and prevent the overfitting. In this regard, we demonstrated that combination of gradient-based learning, the CIM loss, and proposed early-stopping strategy, improves performance relative to the original, MSE-loss based, version of the network (see Table 2). Thus, Tables 1. and 2. show the comparative in-sample and out-of-sample analysis, respectively, for replacing the MSE with CIM, and L-2 with BD regularization.
> 3) *Experimental comparison is not sufficient. Only one method is used for comparison. However, there are many deep learning based clustering methods, they should be used for comparison.*
> - We thank the reviewer for this comment. The motivation for the present paper was upgrading the existing self-supervised network to learn in the truly label-free manner. It is found that combination of gradient-based learning with the CIM loss and early-stopping strategy improved robustness to the presence of the unknown errors in data. In particular, the CIM loss yields smooth learning which enables loss-based label-free early stopping. In this regard, the ablation study was conducted relative to the original (MSE-loss based) version of the network. It is shown in Table 2, that a robust version of the network yielded improved performance on unseen independent test folds. Regarding other deep learning subspace clustering methods, we have pointed out that the majority of them suffer at least from the label-leakage problems, and many of them use the same labels in the training and testing phase. Thus, the comparison would be unfair to the network proposed herein. It is, also, expected that other deep learning based clustering methods, that do not recover subspaces from which samples are generated from, yield better results. That is ‌because simple clustering task is easier than a task of a subspace clustering method. Such comparison would also be unfair.

---

### Decision · Program_Chairs · 2021-01-07
**Final Decision**

**Decision:**

Reject

**Comment:**

The paper proposes a robust formulation of Deep Subspace Clustering (DSC) based on the correntropy induced metric (CIM) of the error. All three reviewers recommend rejection. Their major critiques are limited novelty, insufficient experiments and similar performance to non-deep methods. The rebuttal highlights that the novelty is not DSC or CIM, but rather that the formulation does not require knowing the labels. I agree with the reviewers that the paper's novelty is very limited and didn't find the author's response compelling enough to overturn the reviewer's opinions.